# Tantalum Nitride-Based Theranostic Agent for Photoacoustic Imaging-Guided Photothermal Therapy in the Second NIR Window

**DOI:** 10.3390/nano13111708

**Published:** 2023-05-23

**Authors:** Huixi Yi, Gaoyang Yan, Jinzhen He, Jiani Zhuang, Chengzhi Jin, Dong-Yang Zhang

**Affiliations:** Guangdong Provincial Key Laboratory of Molecular Target & Clinical Pharmacology, National Medical Products Administration, State Key Laboratory of Respiratory Disease, The Fifth Affiliated Hospital, School of Pharmaceutical Sciences, Guangzhou Medical University, Guangzhou 511436, China

**Keywords:** tantalum nitride nanoparticles, photothermal therapy, photoacoustic imaging, theranostics, second near-infrared window

## Abstract

Metal nitrides show excellent photothermal stability and conversion properties, which have the potential for photothermal therapy (PTT) for cancer. Photoacoustic imaging (PAI) is a new non-invasive and non-ionizing biomedical imaging method that can provide real-time guidance for precise cancer treatment. In this work, we develop polyvinylpyrrolidone-functionalized tantalum nitride nanoparticles (defined as TaN-PVP NPs) for PAI-guided PTT of cancer in the second near-infrared (NIR-II) window. The TaN-PVP NPs are obtained by ultrasonic crushing of massive tantalum nitride and further modification by PVP to obtain good dispersion in water. Due to their good absorbance in the NIR-II window, TaN-PVP NPs with good biocompatibility have obvious photothermal conversion performance, realizing efficient tumor elimination by PTT in the NIR-II window. Meanwhile, the excellent PAI and photothermal imaging (PTI) capabilities of TaN-PVP NPs are able to provide monitoring and guidance for the treatment process. These results indicate that TaN-PVP NPs are qualified for cancer photothermal theranostics.

## 1. Introduction

Nanomaterials offer unprecedented opportunities to achieve goals such as promoting precise cancer treatment and alleviating unwanted side effects. In the past, theranostic nanomaterials have been extensively explored to create therapeutic platforms that integrate multiple imaging methods and treatment modalities [1,2,3,4]. As an emerging non-invasive tumor therapy, photothermal therapy (PTT) kills tumors through thermal ablation, which has attracted much attention from researchers around the world [5,6,7,8]. Recently, a large number of photothermal conversion agents have been developed, including gold nanomaterials [9,10,11], carbon-based nanomaterials [12,13,14], metal sulfides [15,16,17], two-dimensional transition metal materials [18,19], organic polymers [20,21], and small molecules [22,23]. However, most nanomaterials are still excited by light sources in the first near-infrared (NIR-I) window with limited penetration depth at present, and their morphological or structural changes are easily affected by their photothermal properties [24,25]. Photothermal conversion materials in the NIR-II window have better tissue penetration depth and weaker tissue absorption. Therefore, it is urgent to develop photothermal conversion agents with good photothermal stability and excitation through the laser in the NIR-II window. In addition, the unique physical and chemical properties of nanomaterials lead to their use as contrast agents for clinical or pre-clinical imaging, such as magnetic resonance imaging [26,27], computed tomography (CT) [28,29], photoacoustic imaging (PAI) [30,31,32], etc. The multifunctional nanoplatforms integration of diagnosis and treatment permits accurate and efficient treatment of tumors [33,34,35]. PAI is a new non-invasive and non-ionizing biomedical imaging method based on ultrasonic signals generated by tissue thermal expansion. It has the advantages of high tissue imaging selectivity and deep penetration depth, which allow real-time guidance for cancer treatment to diagnose disease, guide treatment, and monitor response to treatment. Although many theranostic agents have been developed, there is still a great demand for their development based on PTT in the NIR-II window.

In recent years, some metal nitride nanomaterials, including titanium nitride [36,37] and tungsten nitride [38,39,40], have been developed into photothermal therapeutic agents because of their excellent NIR absorption, photothermal conversion ability, and low toxicity. In addition, tantalum-based nanomaterials with good biosafety and cost effectiveness are also utilized for cancer therapy. For example, their high X-ray absorption due to their high atomic number allows tantalum oxide nanoparticles to be employed as both a CT contrast agent and a radiotherapy sensitizer [41]. Multifunctional tantalum sulfide nanosheets have also been reported to deliver doxorubicin for CT imaging-guided combinatorial photothermal therapy and chemotherapy [42]. Furthermore, tantalum nitride (TaN), one of the important members of tantalum-based compounds, has stable performance and can be used as a precision sheet resistance material or superhard material additive, but its application in the field of biomedicine is less studied. Due to the ideal NIR absorption properties of TaN, it is likely to be used as a PAI-guided PTT agent.

In this work, we developed polyvinylpyrrolidine-functionalized tantalum nitride nanoparticles (TaN-PVP NPs) as theranostic agents for PAI-guided PTT in the NIR-II window. The treatment principle of TaN-PVP NPs was described in Figure 1. The TaN NPs were first acquired by ultrasonic crushing of bulk TaN solids. Furthermore, the as-obtained TaN NPs were modified by the coordination between the metal and carbonyl groups of PVP to improve their dispersibility in water. The morphology and composition of TaN-PVP NPs were characterized by various means. As expected, the as-prepared TaN-PVP NPs exhibited high NIR-II window absorption and excellent photothermal conversion ability, leading to their obvious therapeutic effect on tumor cells. In addition, TaN-PVP NPs could be utilized as a powerful PA contrast agent to monitor in real-time the effective accumulation of NPs in tumor tissues. Moreover, complete tumor elimination was achieved by PAI-guided PTT in the NIR-II window, which had no obvious side effects. These findings confirmed that TaN-PVP NPs had broad application prospects in imaging-guided, accurate NIR-II PTT of cancer.

## 2. Materials and Methods

### 2.1. Materials

TaN and N-methylpyrrolidone (NMP) were purchased from Macklin Reagent (Shanghai, China). PVP was bought from Sigma-Aldrich (St. Louis, MO, USA). Thiazolyl blue (MTT) was acquired from J&K Chemicals (Shanghai, China). The kit of Calcein-AM and Propidium iodide (PI) was bought from Solarbio Life Sciences (Beijing, China). Ultrapure water was obtained from a Milli-Q water purification system (Millipore, Burlington, MA, USA) with a resistivity of 18.2 MΩ/cm.

### 2.2. Synthesis of TaN-PVP NPs

TaN solid was ultrasonic crushed at a power of 300 W (2 s sonication with a 4 s interval) in NMP solution for 24 h, and large pieces of material were removed by centrifugation (3000 rpm). Furthermore, the remaining materials were obtained by centrifugation (10,000 rpm) and washed with water several times to obtain nanoscale TaN. Furthermore, the TaN solution was mixed with the PVP solution and stirred at room temperature overnight. Subsequently, the excess PVP was removed by centrifugation (10,000 rpm) and washed with water several times to obtain PVP-functionalized TaN. The TaN-PVP NPs were stocked at 4 °C for future use.

### 2.3. Characterization

The morphology of nanoparticles was observed by the transmission electron microscope (TEM, FEI Tecnai G2 Spirit, Eindhoven, The Netherlands). The diameter distribution of nanoparticles was measured by counting the size of 50 nanoparticles in TEM images. The sample crystallinity was analyzed by powder X-ray diffraction (XRD, Tokyo, Japan, RigakuD/max-2500 diffractometer) measurement. The hydrodynamic size was measured by dynamic light scattering (DLS, 90Plus/BI-MAS instrument, Brookhaven Instruments Co., New York, NY, USA). X-ray photoelectron spectra (XPS) were carried out on an SSI S-Probe XPS spectrometer with Al Kα radiation as the X-ray source (1486 eV). The optical properties were characterized by the UV/vis-NIR absorption spectra (Cary 60, Agilent Technologies, Santa Clara, CA, USA) and Fourier transform infrared (FT-IR) spectra (L16000300 Spectrum TWO LITA, Llantrisant, UK) measurements, respectively. PAI was performed on a VisualSonics Vevo LAZR system (VisualSonics Inc., New York, NY, USA).

### 2.4. Photothermal and PA Performance of TaN-PVP NPs

To evaluate the photothermal performance of TaN-PVP NPs, various concentrations (100, 200, and 300 μg/mL) of TaN-PVP NP dispersion were irradiated with a 1064 nm laser (0.8 W/cm^2^) for 5 min and imaged by using a thermal imager. The photothermal conversion efficiency of TaN-PVP NPs was calculated by a prior method [43,44].

TaN-PVP NP aqueous dispersions with various concentrations (0, 100, 133, 200, 300, and 400) were filled into plastic pipes to detect their PA signal (Vevo 2100 LAZR system, Toronto, ON, Canada).

### 2.5. Cell Lines and Culture Conditions

4T1 and HEK293T cells were maintained in Dulbecco’s modified eagle medium (DMEM, Gibco BRL, Gaithersburg, MD, USA), which contained 10% FBS (fetal bovine serum, Gibco BRL), 100 μg/mL streptomycin (Gibco BRL), and 100 U/mL penicillin (Gibco BRL). The cells were cultured in a humidified incubator, which provided an atmosphere of 5% CO_2_ and 95% air at a constant temperature of 37 °C.

### 2.6. Animals and Tumor Model

All animal procedures were performed in accordance with the Guidelines for Care and Use of Laboratory Animals of the Guangzhou Medical University, and experiments were approved by the Animal Ethics Committee of the Center of Experiment Animals at the Guangzhou Medical University. Female BALB/c mice were obtained from Guangdong Medical Laboratory Animal Center (Guangzhou, China) with body weights of 19~21 g and housed in stainless steel cages under standard conditions (20 ± 2 °C room temperature, 60 ± 10% relative humidity) with a 12 h light/dark cycle. Mice were injected with 1 × 10^7^ 4T1 cells in 0.1 mL PBS subcutaneously at the right rear flank region. The mice bearing 4T1 tumors with an initial volume of about 100 mm^3^ were used for the animal assay.

### 2.7. Cytotoxicity of TaN-PVP In Vitro

The normal (HEK293T) and 4T1 cells with DMEM medium at a density of 1 × 10^4^ cells/mL were seeded in 96-well plates overnight and then treated with various concentrations (0–200 μg/mL) of TaN-PVP NPs for 20 h. 5 mg/mL of 20 μL of MTT solution was added to each well. The plates were incubated in the incubator for an additional 4 h. The medium was carefully removed, and 150 µL of DMSO was added to each well. The absorbance at 595 nm was measured using a microplate reader. After incubating with TaN-PVP NPs for 4 h, cells in the PTT group were irradiated with a 1064 nm laser at different power densities (0.1–0.8 W/cm^2^) for 5 min. Cell survival rate was calculated by a standard MTT assay according to references [45,46].

The 4T1 cells in DMEM medium at a density of 1 × 10^5^ cells/mL were seeded in 6-well plates overnight and then treated with 200 μg/mL of TaN-PVP NPs. After incubating with TaN-PVP NPs for 4 h, cells in the PTT group were irradiated with a 1064 nm laser at 0.8 W/cm^2^ for 5 min. After staining with calcein AM (2 μM) and PI (4 μM), the live and dead cells were observed under the fluorescence microscope.

### 2.8. In Vivo Imaging and Biodistribution of TaN-PVP NPs

The 4T1-bearing mice were intravenously injected with TaN-PVP NPs (2 mg/mL, 200 μL). The PA signal (Vevo 2100 LAZR system, Toronto, ON, Canada) of tumor tissues was recorded at a series of time points (0, 2, 4, 8, 12, and 24 h).

The 4T1-bearing mice were intravenously injected with TaN-PVP NPs (2 mg/mL, 200 μL). After 24 h, the major organs and tumors were collected after intravenous injection of TaN-PVP NPs. Furthermore, these tissues were digested by aqua regia, and subsequently, the tantalum content was measured by inductively coupled plasma mass spectrometry (ICP-MS) experiments.

The 4T1-bearing mice were intravenously injected with TaN-PVP NPs (2 mg/mL, 200 μL) or phosphate buffer saline (PBS, 200 μL). The temperature changes of tumor tissues were recorded for 5 min after irradiation with a 1064 nm laser at 0.8 W/cm^2^.

### 2.9. In Vivo Photothermal Therapy

4T1 tumor-bearing mice were randomly divided into 4 groups (*n* = 5): (a) PBS, (b) NIR-II, (c) TaN-PVP, and (d) TaN-PVP + NIR-II. The mice from (a) and (b) groups were intravenously injected with PBS (200 μL), while the mice from (c) and (d) groups were intravenously injected with TaN-PVP NPs (2 mg/mL, 200 μL). The mice from (b) and (d) groups were irradiated with a 1064 nm laser (0.8 W/cm^2^) for 5 min at 8 h post-injection. The width and length of tumor tissues and body weight were recorded every 2 days. The tumors from different groups were weighed, photographed, and stained with hematoxylin-eosin (H&E)/ki67 for analysis. The tumor volume was calculated by the following formula: V = w^2^ × L/2, where V is the tumor volume, w is the width of the tumor tissues, and L is the length of the tumor tissues. The tumor section slides were imaged under a microscope.

### 2.10. Hemolysis Assays

The hemolysis rate of TaN-PVP NPs was analyzed by referring to previous articles [43]. Fresh red blood cells were acquired from healthy mice via eyeball blood extraction and washed with PBS. The TaN-PVP NPs were dispersed in PBS at various concentrations (0.125, 0.25, 0.5, 1, and 2 mg/mL). A 1.2 mL sample of dispersion was added to 0.3 mL of the diluted blood and incubated for 4 h at 4 °C. After the solutions were centrifuged at 10,000 rpm for 8 min, the absorbance at 541 nm of the upper solution was detected on a microplate reader. The hemolysis rate of the red blood cells after incubation was calculated according to the following formula: hemolysis rate (%) = (A_sample_ − A_PBS_)/(A_water_ − A_PBS_) × 100%, where A_water_, A_PBS_, and A_sample_ were the 541 nm absorbance values of the H_2_O group, the PBS group, and the sample groups, respectively.

### 2.11. Detection of Kidney/Liver Indicators

Mice blood was obtained after 14 days of treatment, and serum was collected by centrifugation. The levels of blood urea nitrogen (BUN), aspartate, transaminase (AST), creatinine (CREA), and alanine transaminase (ALT) in serum from the indicated groups were determined.

### 2.12. Statistical Analysis

Representative results were depicted in this report, and the data were presented as means ± standard deviations (SD). **** *p* < 0.0001 was considered a statistically significant difference between groups.

## 3. Results and Discussion

### 3.1. Synthesis and Characterization of TaN-PVP NPs

First, the TEM images revealed that the as-synthesized TaN-PVP NPs appeared to have random morphologies due to ultrasonic breakage, ranging in size from 100 to 200 nm (Figure 1A). TaN-PVP NPs were well-dispersed in water, and their hydrophilic diameter was measured to be about 190 nm (Figure 1B). Meanwhile, the C=O stretching vibrations at 1650 cm^−1^ found in the FT-IR spectrum of TaN-PVP NPs indicated the successful modification of PVP (Appendix A) [47]. In addition, XRD and XPS were conducted to investigate the chemical composition of the as-prepared TaN-PVP NPs. XRD results confirmed the crystalline structure of TaN as shown in Figure 1C, which was consistent with hexagonal TaN (JCPDS No. 39-1485) and Ta_2_N phases (JCPDS No. 26-0985). Moreover, the characteristic peaks of N 1s at 404 eV and Ta 4f at 25.7 eV were observed in the XPS spectra of TaN-PVP NPs, as presented in Figure 1D,E. Importantly, TaN-PVP NPs presented a broad absorbance in the NIR-I and NIR-II windows (Figure 1F), suggesting the NIR-II PTT and PAI potential of TaN-PVP.

### 3.2. Photothermal Property and PAI of TaN-PVP NPs

The high NIR-II absorption capacity of TaN-PVP NPs encourages us to study their photothermal performances, which play a vital role in the applications of PTT. The photothermal performances of TaN-PVP NPs aqueous dispersions with various concentrations (0, 100, 200, and 300 μg/mL) were then investigated by using a 1064 nm laser. The increase in the temperature of the solution depended on the irradiation duration and nanomaterial concentration (Figure 2A,B). The temperature of the TaN-PVP NPs solution at 300 μg/mL increased from 28.0 °C to 56.4 °C after 5 min of irradiation by a 1064 nm laser at a power density of 0.8 W/cm^2^, while the temperature of ultrapure water only increased by 2.5 °C under the same conditions, demonstrating the excellent photothermal property of TaN-PVP NPs. Meanwhile, the photothermal conversion efficiency of TaN-PVP NPs was measured as 33.1% based on the prior method (Figure 2C,D) [43,44]. The investigation of the photostability of TaN-PVP NPs was also conducted; the temperature change of the TaN-PVP NP dispersion did not change significantly under the condition of repeated irradiation (Figure 2E), and the absorption of the solutions did not decrease obviously (Appendix A), which indicated the good photothermal stability of TaN-PVP NPs. Due to their high photothermal conversion efficiency, the TaN-PVP NPs possess the potential to be used as PA contrast agents. The intensity of the PA signal was positively correlated with the concentration of the TaN-PVP NPs solutions (Figure 2F), suggesting that it could be utilized for subsequent PAI.

### 3.3. PTT Efficiency of TaN-PVP NPs In Vitro

The cytotoxicity of TaN-PVP NPs was determined by the standard MTT assay using normal (HEK293T) and cancer (4T1) cells [45,46]. No apparent cytotoxicity was found on both HEK293T and 4T1 cells after treatment with TaN-PVP at the indicated concentrations (Figure 3A,B), indicating the good biocompatibility of the TaN-PVP NPs for biomedical applications. In contrast, the cell survival rate was lower than 10% in 4T1 cells incubated with 200 μg/mL of TaN-PVP with irradiation (Figure 3B). To evaluate the in vitro treatment of TaN-PVP NPs with or without NIR laser exposure, live/dead staining were further carried out. As presented in Figure 3C, the red fluorescence (dead cell) was found in abundance in the PTT group but almost none in the other groups. Additionally, by changing the power density of the laser, it was found that cell survival rate was closely related to light density. Cell mortality increased with the increase in optical power density (Figure 3D). These results confirmed that TaN-PVP could effectively kill tumor cells by heating them under NIR-II laser irradiation.

### 3.4. Biodistribution and Imaging of TaN-PVP NPs In Vivo

Precise monitoring of drugs in vivo is particularly needed because it may open up a new avenue for directing treatment processes, monitoring treatment responses, and avoiding damage to surrounding healthy tissue caused by external laser radiation, thereby reducing associated side effects. PAI based on the PA effect caused by NIR absorption and subsequent thermal expansion provides higher spatial resolution of soft tissues, which can be used for real-time monitoring. Hence, the in vivo PA signals of TaN-PVP were further collected on the 4T1 tumor-bearing mice after intravenous administration with TaN-PVP. The PA signals of tumor tissues reached maximum at 8 h post-injection (Figure 4A,B), indicating the efficient accumulation of TaN-PVP due to the EPR effect. Biodistribution analysis also verified the effective accumulation in tumor tissues (5.6% ID/g) of TaN-PVP (Figure 4C) as measured by ICP-MS analysis. Moreover, the temperature of the tumor tissues was recorded in real time by using a thermal imager. These results showed that the tumor tissue of mice in the TaN-PVP group increased by 24.5 °C, whereas that of the control group increased by only 7.8 °C (Figure 4D,E). These studies revealed that TaN-PVP NPs could be used as efficient contrast agents for PAI and to guide PTT.

### 3.5. PTT Effect of TaN-PVP NPs In Vivo

Inspired by the therapeutic effect of NPs in vitro, we carried out mouse experiments to assess in vivo anticancer efficacy. The mice bearing 4T1 tumors with an initial volume of about 100 mm^3^ were first randomly divided into the four indicated groups. The mice were photographed before and after treatment, and the tumors were collected at the end of treatment. The tumor photographs and tumor weights for each group were recorded. As shown in Figure 5A–D, the tumors were completely eliminated in the NIR-II PTT group, while the NIR-II and TaN-PVP groups had no significant inhibition compared with the control (PBS) group. Meanwhile, the H&E staining results showed that the cell density of tumor tissues in the PTT group decreased significantly compared with the control group (Figure 5E), indicating the killing effect of nanomaterials. Interestingly, the proliferative capacity of tumor cells was also seen to be significantly decreased in the PTT group as measured by ki67 staining (Figure 5F) [48,49]. All these data confirmed that TaN-PVP had an excellent photothermal therapeutic effect, which allowed it to completely eliminate tumors in vivo. These findings were consistent with the results of the in vitro experiment.

### 3.6. Biocompatibility Evaluation of TaN-PVP NPs

The safety of nanomaterials is an essential concern when it comes to the design of nanomaterials for biomedical applications. The biocompatibility of TaN-PVP NPs was first studied based on hemolysis assays, liver/kidney indicator analyses, and major organ staining analyses. There was no significant difference in the weight of the mice after treatment (Figure 6A). The hemolysis ratio of red blood cells was lower than 5% when incubated with TaN-PVP NPs at the indicated concentrations (Appendix A). At the same time, the renal/liver index levels of BUN, AST, CREA, and ALT were not obviously abnormal in 4 groups (Figure 6B,C, Appendix A and Appendix A), suggesting no noticeable renal and hepatic dysfunction induced by TaN-PVP NPs. Moreover, no inflammation or damage was discovered in the major organs in all groups (Figure 6D). These results verified that TaN-PVP NPs with good biosafety could be considered candidates for PAI-guided PTT.

## 4. Conclusions

In summary, we had successfully prepared a TaN-PVP nanoplatform via an ultrasonic crushing method, which had the potential to realize PAI-guided PTT as an anticancer theranostic agent. The as-synthesized TaN-PVP NPs with good dispersion not only had high photothermal conversion ability but also possessed excellent photothermal stability. In addition, TaN-PVP NPs were highly efficient in photothermal ablation of tumor cells with low dark cytotoxicity. Moreover, the PAI capability of TaN-PVP NPs provided the ability to precisely guide the PTT under the irradiation of the NIR-II laser, leading to the complete elimination of the tumor. Importantly, the good biocompatibility of TaN-PVP NPs made their further medical application possible. Overall, the TaN-based theranostic nanoplatform offered a novel candidate for the development of precise and efficient theranostic agents.

## Data Availability

The data presented in this study are available on request from the corresponding author.

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
