# Peer review of "Tantalum Nitride-Based Theranostic Agent for Photoacoustic Imaging-Guided Photothermal Therapy in the Second NIR Window"

_nanomaterials, 2023, doi:10.3390/nano13111708_

Round 1

Reviewer 1 Report

This manuscript (Yi, H, et al) is reporting a tantalum nitride (TaN)-based nanomaterial for photoacoustic imaging (PAI) guided-photo thermal therapy. It describes preparation of PVP-coated TaN and characterization of its physicochemical properties such as by TEM and DLS, and it demonstrated ability for photo thermal activation using NIR-II light. Its authors showed this nanomaterial is effective for generating photo thermal images in solutions as well as in a tumor model in mice. They also showed it is biocompatible however it is activatable by NIR-II light, which results in induction of potent tumor cell death in vitro as well as killing tumors in mice.

Overall, this manuscript is written well. It offers a detailed description of TaN investigated in vitro and in vivo. While such systems have been reported in literature, this manuscript offers clear demonstrations of its PTT and PAI capability. This manuscript is recommended for publication in Nanomaterials. Few minor comments are provided below for consideration during its revision.

Comments:

1.     Fig 2a and 2b. In their legends, provide sample concentrations used in their PTT tests.

2.     Please provide a statement of an institutional approval on animal care and studies.

3.     Typos: “cruves .. (Fig 4E)

In addition to typos noted in the comment section, please pay attention to other typos and errors.

Author Response

Reviewer #1: This manuscript (Yi, H, et al) is reporting a tantalum nitride (TaN)-based nanomaterial for photoacoustic imaging (PAI) guided-photo thermal therapy. It describes preparation of PVP-coated TaN and characterization of its physicochemical properties such as by TEM and DLS, and it demonstrated ability for photo thermal activation using NIR-II light. Its authors showed this nanomaterial is effective for generating photo thermal images in solutions as well as in a tumor model in mice. They also showed it is biocompatible however it is activatable by NIR-II light, which results in induction of potent tumor cell death in vitro as well as killing tumors in mice.

Overall, this manuscript is written well. It offers a detailed description of TaN investigated in vitro and in vivo. While such systems have been reported in literature, this manuscript offers clear demonstrations of its PTT and PAI capability. This manuscript is recommended for publication in Nanomaterials. Few minor comments are provided below for consideration during its revision.

Response: Thanks for the reviewer’s positive comment.

Comment 1:  Fig 2a and 2b. In their legends, provide sample concentrations used in their PTT tests.

Response: The sample concentrations used in their PTT tests was added in the legends of Fig. 2A and 2B. “(100-300 μg/mL)”

Comment 2: Please provide a statement of an institutional approval on animal care and studies.

Response: A statement of an institutional approval on animal care and studies were supplemented in “2.6 Animals and Tumor Model”. “All animal procedures were performed in accordance with the Guidelines for Care and Use of Laboratory Animals of the Guangzhou Medical University and experiments were approved by the Animal Ethics Committee of the Center of Experiment Animals at Guangzhou Medical University.”

Comment 3: Typos: “cruves .. (Fig 4E)

Response: We have revised it.

Reviewer 2 Report

This is an interesting paper on the use of specific nanoparticles in theranostics. Number of such papers accumulate in the literature enormously, however, the idea presented by Authors is worth to be published. In my opinion in order that paper could be published some remerks should be taken under consideration. They are as follows:

1./ Authors intensively use abbreviations. It would be beneficial to work to add a list of them;

2./ Authors please comment how the distribution of diameters of the nanoparticles was measured (citation is welcome);

3./  All the citations are done to Chinese Authors (with one exception). I do know that the choise of citationw is a priviledge of Authors, but from the paper it looks like photothermal therapy and theranostics had been discovered and used in China exclusively. This is irresponsible.

Some small comments are as follows (in the order of appearance in the manuscript):

1./ Authors please chcek English, especially use of singulars versus plurals;

2./ first sentence in paragraph 2.7. is unclear;

3./ Remove first sentence in paragraph 3.1.;

4./ cell survival rate is lower (not less) than ... (page5 line 168);

5./ there is wrong numbering of Figure in paragraph 3.4. - should be Figure 4 not 3;

6./ caption to (C) in Figure 4 is unclear;

7./ remove "indicated" (page 7 line 199);

8./ should be "considered as candidate for... (end of page 8, line 227);

9./ add an explanation (hemathoxylin) after H&E in captionto Figure 6.

English is o.k. and only some improvement would be acknowledged.

Author Response

Reviewer #2: This is an interesting paper on the use of specific nanoparticles in theranostics. Number of such papers accumulate in the literature enormously, however, the idea presented by Authors is worth to be published. In my opinion in order that paper could be published some remerks should be taken under consideration.

Response: Thanks for the reviewer’s suggestion.

Comment 1: Authors intensively use abbreviations. It would be beneficial to work to add a list of them.

Response: We have added the abbreviations in the manuscript.

Abbreviations: photothermal therapy (PTT), first near infrared (NIR-I), second near infrared (NIR-II), computed tomography (CT), photoacoustic imaging (PAI), tantalum nitride (TaN), polyvinylpyrrolidine-functionalized tantalum nitride nanoparticles (TaN-PVP NPs), N-methylpyrrolidone (NMP), Thiazolyl blue (MTT), Propidium iodide (PI), transmission electron microscope (TEM), X-ray diffraction (XRD), dynamic light scattering (DLS), X-ray photoelectron spectra (XPS), Fourier transform infrared (FT-IR), dulbecco's modified eagle medium (DMEM), phosphate buffer saline (PBS), hematoxylin-eosin (H&E), blood urea nitrogen (BUN), aspartate, transaminase (AST), creatinine (CREA), alanine transaminase (ALT), standard deviations (SD).”

Comment 2: Authors please comment how the distribution of diameters of the nanoparticles was measured (citation is welcome).

Response: The method of diameter distribution of nanoparticles was added to the manuscript. “The diameter distribution of nanoparticles was measured by counting the size of 50 nanoparticles in TEM images.”

Comment 3: All the citations are done to Chinese Authors (with one exception). I do know that the choise of citationw is a priviledge of Authors, but from the paper it looks like photothermal therapy and theranostics had been discovered and used in China exclusively. This is irresponsible.

Response: Thanks especially for this suggestions. We replaced and added some references on photothermal therapy and theranostics by non-Chinese authors. At the same time, some of the literature was written by scholars from Singapore or co-authored with Chinese scholars.

Some small comments are as follows (in the order of appearance in the manuscript)

Comment 4: Authors please chcek English, especially use of singulars versus plurals.

Response: We have examined the language, including the use of singles and plurals.

Comment 5: first sentence in paragraph 2.7. is unclear.

Response: The experimental methods in paragraph 2.7. were supplemented in detail.

“The hemolysis rate of TaN-PVP NPs was analyzed by referring to previous articles [40]. Fresh red blood cells was acquired from healthy mice via eyeball blood extraction, and washed with PBS. The TaN-PVP NPs were dispersed in PBS with various concentrations (0.125, 0.25, 0.5, 1, and 2 mg/mL). 1.2 mL sample dispersion was added to 0.3 mL of the diluted blood and incubated for 4 h at 4 °C. After the solutions were centrifuged at 10000 rpm for 8 min, the absorbance at 541 nm of the upper solution was detected on a microplate reader. The hemolysis rate of the red blood cells after incubation was calculated according to the following formula: hemolysis rate (%) = (Asample - APBS)/(Awater - APBS) × 100%, where Awater, APBS, and Asample were the 541 nm absorbance values of the H2O group, the PBS group, and the sample groups, respectively.”

Comment 6: Remove first sentence in paragraph 3.1.

Response: The first sentence in paragraph 3.1 were removed.

Comment 7: cell survival rate is lower (not less) than ... (page5 line 168).

Response: We have revised it.

Comment 8: there is wrong numbering of Figure in paragraph 3.4. - should be Figure 4 not 3.

Response: We have revised it.

Comment 9: caption to (C) in Figure 4 is unclear.

Response: The caption to (C) in Figure 4 were supplemented in detail.

Comment 10: remove "indicated" (page 7 line 199).

Response: The "indicated" on page 7 line 199 were removed.

“Biodistribution (major organs and tumor tissues) of Ta element in 4T1-tumor-bearing mice at 24 h post-injection of TaN-PVP as determined by ICP-MS.”

Comment 11: should be "considered as candidate for... (end of page 8, line 227).

Response: We have revised it. “These results verified that TaN-PVP NPs with good biosafety could be considered as candidates for PAI guided PTT.”

Comment 12: add an explanation (hemathoxylin) after H&E in captionto Figure 6.

Response: We have supplemented the full name of hematoxylin-eosin (H&E) in the manuscript.

Reviewer 3 Report

The authors synthesized polyvinylpyrrolidone functionalized tantalum nitride nanoparticles (defined as TaN-PVP NPs) for photoacoustic 12 imaging (PAI) guided PTT of cancer in second near-infrared (NIR-II) window. They characterized the synthesized nanoparticles using TEM, XRD, XPS, UV-visible absorption, in vitro cellular toxicity, and PT.

(1)   Figure 1 caption (F) is missing or mis-ordered.

(2)   TaN nanoparticles were mixed overnight to obtain PVP-coated TaN nanoparticles. could you explain that just mixing provided PVP-coated TaN nanoparticles ?

(3)   Could you discuss the nanoparticle morphology in the text because TEM image showed non-spherical particles ?

Author Response

Reviewer #3: The authors synthesized polyvinylpyrrolidone functionalized tantalum nitride nanoparticles (defined as TaN-PVP NPs) for photoacoustic imaging (PAI) guided PTT of cancer in second near-infrared (NIR-II) window. They characterized the synthesized nanoparticles using TEM, XRD, XPS, UV-visible absorption, in vitro cellular toxicity, and PT.

Response: Thanks for the reviewer’s positive comment.

Comment 1: Figure 1 caption (F) is missing or mis-ordered.

Response: We have revised the caption of Figure 1.

Comment 2: TaN nanoparticles were mixed overnight to obtain PVP-coated TaN nanoparticles. could you explain that just mixing provided PVP-coated TaN nanoparticles?

Response: Due to the coordination between metal and carbonyl group of PVP, the TaN NPs were successfully modified by mixing PVP and TaN. The corresponding explanation has been added to the manuscript. “the as-obtained TaN NPs was modified by the coordination between metal and carbonyl group of PVP to improve their dispersibility in water.”

Comment 3: Could you discuss the nanoparticle morphology in the text because TEM image showed non-spherical particles?

Response: We have supplemented the discussion of nanoparticle morphology in the manuscript. “The TEM images revealed the as-synthesized TaN-PVP NPs appeared to be random morphologies due to ultrasonic breakage, ranging in size from 100 to 200 nm.”